# FEDERATED SELF-SUPERVISED LEARNING FOR HETEROGENEOUS CLIENTS

## ABSTRACT

Federated Learning has become an important learning paradigm due to its privacy and computational benefits. As the field advances, two key challenges that still remain to be addressed are: (1) system heterogeneity - variability in the compute and/or data resources present on each client, and (2) lack of labeled data in certain federated settings. Several recent developments have tried to overcome these challenges independently. In this work, we propose a unified and systematic framework, *Heterogeneous Self-supervised Federated Learning* (Hetero-SSFL) for enabling self-supervised learning with federation on heterogeneous clients. The proposed framework allows collaborative representation learning across all the clients without imposing architectural constraints or requiring presence of labeled data. The key idea in Hetero-SSFL is to let each client train its unique self-supervised model and enable the joint learning across clients by aligning the lower dimensional representations on a common dataset. The entire training procedure could be viewed as self and peer-supervised as both the local training and the alignment procedures do not require presence of any labeled data. As in conventional self-supervised learning, the obtained client models are task independent and can be used for varied end-tasks. We provide a convergence guarantee of the proposed framework for non-convex objectives in heterogeneous settings and also empirically demonstrate that our proposed approach outperforms the state of the art methods by a significant margin.

## 1 INTRODUCTION

Federated learning has become an important learning paradigm for training algorithms in a privacy preserving way and has gained a lot of interest in the recent past. While traditional federated learning is capable of learning high performing models (Li et al., 2021b), two practical challenges that remain under studied are: system heterogeneity, and lack of labeled data. In several real world scenarios, the clients involved in the training process are highly heterogeneous in terms of their data and compute resources. Requiring each client to train identical models, like in traditional FL, may thus be severely limiting. Similarly, assuming each client's local data resources to be fully labeled may not be pragmatic as annotating data is time consuming and may require expertise. Our focus in this work is to jointly address these two challenges in a systematic way so as to substantially increase the scope of FL approaches. Prior works have studied these two issues separately, and the approaches taken do not offer a natural way to be combined so as to provide an effective heterogeneous self-supervised FL framework. For instance, to alleviate the scarcity of labeled data on local clients, both semi-supervised learning (Zhang et al., 2021b; Jeong et al., 2021; Lu et al., 2022; Lubana et al., 2022) and self-supervised learning (Zhuang et al., 2021; 2022; He et al., 2022; van Berlo et al., 2020) methods have been proposed, but they all assume identical model architectures on each client and in fact do not extend to heterogeneous settings. Some aspects of system heterogeneity have been independently addressed for supervised learning by building personalised models on clients (Tan et al., 2022; Jiang et al., 2020; Fallah et al., 2020), but still assuming identical architectures. Existing recent works on federated learning with independent architectures across clients consider standard supervised learning scenarios (Makhija et al., 2022) rather than self-supervised learning.

The need for self-supervised FL arises in multiple applications. For example, consider cross-silo analytics in healthcare systems where different hospitals may possess varying amounts of private medical images. Here, the data can neither be centralised nor can undergo extensive annotations. Moreover, to expect each client (e.g., a hospital) to train local models of identical capacities can

be highly inhibiting. In such cases, the smaller capacity clients which are incapable of training large models of common architecture will not be accommodated in federated learning. On the other hand, if the common model architecture size is reduced, some of the clients will not be fully engaged. Differences in compute resources and the learning capacity demand system heterogeneity. To the best of our knowledge, ours is the first work that proposes a general framework for federated self-supervised learning in heterogeneous settings where non-identical clients can implement distinct, independent model architectures and obtain personalised solutions.

The proposed framework is novel and flexible, and allows clients to train self-supervised models with unique (locally tuned) structures while still using the learnings from other clients in the network, in an architecture agnostic way. To achieve this, we add a proximal term to each client's loss function that helps in aligning the learnt lower dimensional representations (aka embeddings) across clients. This way of transferring and acquiring global knowledge allows variability in client model architectures while keeping the entire learning process independent of labeled training data. Furthermore, we use a kernel based distance metric for proximity calculation which provides much more flexibility to the clients in defining their own lower dimensional space without any constraints.

**Our Contributions** are summarized as follows :

1. Our main contribution is the new framework, *Hetero-SSFL*, for training heterogeneous models in a federated setting in an unsupervised way. Hetero-SSFL allows each client to train its own customized model, using local data and computing resources while also utilising unlabeled supervision from peers.

2. We perform thorough experimental evaluation of the proposed approach in both image-based and text-based self-supervised learning settings. We observe that the proposed flexible approach substantially improves the accuracy of unsupervised and self-supervised federated learning in heterogeneous settings.

3. We also provide theoretical analysis and convergence guarantee of the algorithm for non-convex loss functions.

**Organization.** The rest of the paper is organised as follows. Section 2 provides a brief background on Federated Learning, Self-supervised Learning and related developments. In Section 3, we go over the preliminaries and then propose our framework. We study the convergence guarantee of the algorithm in Section 4 and include the related proofs in the Appendix. A thorough experimental evaluation of our method on different types of datasets is presented in Section 5 and we conclude the paper in Section 6.

## 2 RELATED WORK

This section provides an overview of the most relevant prior work in the fields of federated learning, self-supervised learning and federated self-supervised learning.

### 2.1 FEDERATED LEARNING(FL)

The problem of training machine learning models in distributed environments with restrictions on data/model sharing was studied by several researchers in the data mining community in the early 2000s under titles such as (privacy preserving) distributed data mining (Kargupta & Park, 2000; Gan et al., 2017; Aggarwal & Yu, 2008). Much of this work was for specific procedures such as distributed clustering (Merugu & Ghosh, Nov, 2003) including in heterogenous settings with different feature spaces (Merugu & Ghosh, 2005), and distributed PCA (Kargupta et al., 2001), or for specific models such as SVMs (Yu et al., 2006). Subsequently, with the re-surfacing in popularity of neural networks and proliferation of powerful deep learning approaches, the term "Federated Learning" got coined and popularized largely by an influential paper (McMahan et al., 2017) which introduced FedAvg. Indeed FedAvg is now considered the standard distributed training method for Federated Learning, which has become a very active area of research since then. One of the key challenges in this setting is the presence of non-iid datasets across clients. Several modifications of the original FedAvg algorithm have been proposed to address this challenge. Some of these approaches focus on finding better solutions to the optimization problem to prevent the divergence of the global solution (Li et al., 2020; Karimireddy et al., 2020; Zhang et al., 2021a; Pathak & Wainwright, 2020; Acar et al., 2021; Karimireddy et al., 2021) whereas some suggest the modification of the training procedure to incorporate appropriate aggregation of the local models (Chen & Chao, 2021; Wang et al., 2020a;

Yurochkin et al., 2019; Wang et al., 2020b; Singh & Jaggi, 2020). Creating local personalised models for each client to improve the overall method is also well researched (Collins et al., 2021; Ghosh et al., 2020; Smith et al., 2017a; Li et al., 2021b; Smith et al., 2017b; Sattler et al., 2021; Yu et al., 2020). Also, augmenting the data distributions by creating additional data instances for learning (Hao et al., 2021; Goetz & Tewari, 2020; Luo et al., 2021) has seen to help with the collaborative effort in federated learning.

## 2.2 SELF-SUPERVISED LEARNING(SSL)

SSL aims at training high performing complex networks without the requirement of large labeled datasets. It does not require explicitly labeled data and target pairs but instead uses inherent structure in the data for representation learning. The two types of SSL methods include contrastive based and non-contrastive based SSL. The contrastive methods work on triplets of the form $(x, x', y)$ where $x$ and $x'$ are different views of the same instance and $y$ is a non-compatible data instance. The key idea is to train models by adjusting the parameters in such a way that the representations(embeddings) obtained for $x$ and $x'$ are nearly identical and that of $x$ and $y$ are unalike. Since obtaining these triplets from the dataset does not necessarily require presence of the labels, for example for images, $x$ and $x'$ could be different augmentations of the same image and $y$ could be an image of a different object, these methods could learn from large volume of data. Recent works in this field suggest many different ways of generating the data triplets for training (Chen et al., 2020a; He et al., 2020; Wu et al., 2018; Yan et al., 2020). Non-contrastive methods, on the other hand, work without generating explicit triplets. One form of the non-contrastive methods use clustering based methods to form a group of objects and use the cluster assignments as pseudo-labels for training the model (Caron et al., 2018; 2020). Another form of non-contrastive methods use joint embedding architectures like Siamese network but keep one of the architectures to provide target embeddings for training (Chen & He, 2020; Grill et al., 2020; Chen et al., 2020b).

## 2.3 FEDERATED SELF-SUPERVISED LEARNING

The interest in self-supervised federated learning is relatively new. Although MOON (Li et al., 2021a) proposed using a contrastive loss in supervised federated learning, it was FedU (Zhuang et al., 2021) that proposed a way to leverage unlabeled data in a decentralised setting. Subsequently, two parallel works (Zhuang et al., 2022), (He et al., 2022) proposed frameworks for training self-supervised models in a federated setting and compared different self-supervised models. In all of these approaches though, the clients train local models of identical architectures. This common architecture is communicated by the server at the beginning of the training. In each round, the clients perform update steps on the local models and send the local models to the central server at the end of the round. The server appropriately aggregates the client models to generate the global model for that round and communicates the global model parameters back to each client. The prominent idea in these latest developments is the divergence aware updates of the local model to the global model wherein, if the local models are very different from the global model, the local models are not updated with the global model, which can be seen as an initial step towards creating more personalised models for each client. On the other hand, (Shi et al., 2021) does explore heterogeneous architectures across clients but uses them to train a common global model which again does not suit heterogeneity. Additionally, some very recent works have explored self-supervised FL for specific applications like acoustics, human-activity recognition etc. (Feng et al., 2022; Ek et al., 2022; Rehman et al., 2022).

## 3 METHODOLOGY

In order to enable federated learning for heterogeneous clients in an unsupervised way, we propose a new framework using self-supervised federated learning and name it Hetero-SSFL. This framework is unique as it allows heterogeneous clients to participate in the training process, exhibits high empirical performance and is guaranteed to converge. In this section we first formally describe the problem setting and then elaborate on our proposed solution.

### 3.1 PROBLEM DEFINITION

A federated learning setting consists of a set of clients $k \in |N|$, with each client $k$ having local data instances $\mathcal{X}_k$ of size $n_k$ drawn from the local data distribution $\boldsymbol{D}_k$. The local objective at $k^{th}$ client is to solve for $\min_{\mathcal{W}_k} \mathcal{L}_k = \ell(\mathcal{W}_k; \mathcal{X}_k)$ where $\ell(.)$ is a suitable loss function. After the local optimization, the federation part of the learning allows the server to access all the clients' learnings in the form of the learned set of weights $\mathcal{W}_1, \mathcal{W}_2, \ldots, \mathcal{W}_N$ and utilise it for achieving global objective.

In traditional FL methods, it is assumed that all $\mathcal{W}_k$'s are identical in shape and in each round the server combines the local models to learn a global model $\hat{\mathcal{W}}$ of the form $\hat{\mathcal{W}} = g(\mathcal{W}_1, \mathcal{W}_2, \ldots, \mathcal{W}_N)$, where $g$ is an appropriate aggregation function, and sends $\hat{\mathcal{W}}$ back to the clients for further updates. In contrast, our solution does not require the local models (and thus the local weights) to be of identical size, as we deploy a novel and different way of utilizing the global knowledge, $\mathcal{W}_1, \mathcal{W}_2, \ldots, \mathcal{W}_N$, to achieve collaboration for each client's benefit.

## 3.2 HETERO-SSFL FRAMEWORK

We introduce the Hetero-SSFL framework in this section and provide details of all the components in the framework here. The end-to-end pipeline is depicted in Figure 1 and the algorithm is detailed in Algorithm 1.

**General Formulation** As mentioned above, the federated self-supervised learning problem involves each client $k$ solving a local optimization problem, $\min_{\mathcal{W}_k} \mathcal{L}_k = \ell(\mathcal{W}_k)$, for some self-supervised loss, $\ell(\mathcal{W}_k)$. In Hetero-SSFL, we modify the loss function for each client to the following:

$$\min_{\mathcal{W}_k} \mathcal{L}_k = \ell(\mathcal{W}_k) + \mu \mathrm{d}\left( \Phi_k(\mathbf{X}; \mathcal{W}_k), \bar{\Phi}(\mathbf{X}; (t-1)) \right). \tag{1}$$

where $\mathrm{d}(.,.)$ is a suitable distance metric, $\Phi_k(.;.)$ is used to denote the representation learning component of the $k^{th}$ client, $\Phi_k(\mathbf{X}; \mathcal{W}_k)$ denotes the representation matrix of $k^{th}$ client under weights $\mathcal{W}_k$ on dataset $\mathbf{X}$ and $\bar{\Phi}$ denotes the aggregated representations of all clients and is computed as

$$\bar{\Phi}(\mathbf{X}; (t)) = \sum_{j=1}^{N} w_j \Phi_j(\mathbf{X}; \mathcal{W}_j(t)). \tag{2}$$

Here, $w_j$ is the weight given to the representations obtained from the $j^{th}$ client. The client weight $w_j$ could be kept higher for clients with larger resources (clients which are capable of training larger capacity models and obtain better representations), or could be set to learn in the training pipeline. The new loss function enables collaboration across clients by aligning each client's representations and simultaneously guiding the training on all clients. Specifically, we modify the local loss function at each client to contain a proximal term that measures the distance between the local representations and the representations obtained on all other clients.

This framework could be used for any self-supervised learning application by suitably choosing the self-supervised loss function $\ell(\mathcal{W}_k)$, the distance function $\mathrm{d}(.,.)$, and the network architecture $\Phi_k(.;.)$. In this section we present the details of self-supervised learning for image recognition tasks and provide an extension for text classification task in Appendix B, thereby providing two concrete instances of this general formulation.

**Local Training** We assume each client contains unlabeled datasets and employ self-supervised learning on each client. While there are many ways to achieve self-supervision, in this work we focus on non-contrastive methods like BYOL (Grill et al., 2020) and SimSiam (Chen & He, 2020) which involve two embedding architectures like Siamese networks for achieving self-supervision. The two parallel networks are called the *online* network and *target* network respectively. The target network's task is to provide targets for regression to the online network whereas the online network is trained to learn representations. The target network consists of an encoder and the online network is composed of an encoder and a prediction layer on top of it, whose role is to learn better representations under self-supervised loss. The weights of the target network are just a moving average of the weights of the online network and the target network can thus be thought of as a stable version of the online network. In federated self-supervised learning with these networks, the local set of parameters for each client $k$ include $\mathcal{W}_k = \{W_k^o, W_k^p, W_k^t\}$ where $W_k^o$ parameterises the representation learning component of the online network with $W_k^p$ being the prediction layer and $W_k^t$ is the target network. The local loss at each client $\ell(.)$ is similar to self-supervised loss used in central self-supervised learning and is given by -

$$\ell(\mathcal{W}_k) = ||\mathcal{F}(v'; [W_k^o, W_k^p]) - \mathcal{F}'(v''; W_k^t)||^2,$$

i.e., it is the mean squared error between the outputs of the online network and the target network on different views(augmentations), $v'$ and $v''$, of the same image $v$. $\mathcal{F}(,;.)$ and $\mathcal{F}'(,;.)$ denote the online and target network functions respectively. The representations $\Phi(\mathbf{X}; .)$ are obtained from the output of the client's online network under parameters $\mathcal{W}^o = [W^o, W^p]$ at a layer suitable for

defining the representation space, and this could potentially be different from $\mathcal{F}$ which denotes the entire online network. In our framework, we use the output of the predictor layer for an instance $x$ as $\Phi(x; .)$.

**Communication with the Server** Different from traditional FL where each client sends the local models to server and the server creates a global model by performing element-wise aggregation on the local model weights, we gather representations at the server. We assume that the server has access to an unlabeled set of data points (which could be obtained in practice from open sources like images from web) and call it *Representation Alignment Dataset* (RAD). The RAD is a publicly available dataset used for the purpose of aligning representations and doesn't possess any special properties. $\Phi_i(\mathbf{X}; \mathcal{W}_i)$ is thus a matrix of dimensions $L$ x $d_i$ where $L$ is the number of data points in $\mathbf{X}$ and $d_i$ is the size of embedding for each point. In each global training round $t$, server sends the RAD to all the clients. All the clients use the RAD, $\mathbf{X}$, in the loss function as described above while simultaneously training local models. After the clients finish local epochs, the final representations obtained for RAD $\bar{X}$ on each client, $\Phi_i(\mathbf{X}; \mathcal{W}_i)$, are sent to the server. The server then aggregates the local representations from all clients to form the global representation matrix $\bar{\Phi}(\mathbf{X}; (t))$, as in Equation 2, which is then again sent to the clients for training in the next round. *This methodology is similar to what (Makhija et al., 2022) does, but for supervised learning. Here we tackle the problem of lack of labelled data by rather doing federated self-supervised learning.*

### 3.3 THE PROXIMAL TERM

In this sub-section we describe the second part of the loss function and provide details on the proximal term and the distance function.

The role of the proximal term is to compare the representations obtained from various neural networks. This requires the distance function to be able to meaningfully capture the similarity in the high dimensional representational space. Moreover, the distance function should be such that it is sensitive to the functional behavior(e.g., outputs from initial layer versus penultimate layer) and does not change as much with the inconsequential modifications like change in random initializations. The types of distance functions used in the literature for comparing neural network representations include Canonical Correlation Analysis(CCA), Centered Kernel ALignment(CKA) and Procustres distance based measures (Ding et al., 2021; Kornblith et al., 2019). While the Procustres distance based metrics cannot compare the representations obtained in spaces of different dimensions, the CCA based metrics are not suitable for training through backpropagation. In contrast, properties of the CKA metric like ability to learn similarities between layers of different widths, invariance to random initializations, invertible linear transformations and orthogonal transformations lend it useful for use in the proximal term.

CKA measures the distance between the objects in different representation spaces by creating similarity(kernel) matrices in the individual space and then comparing these similarity matrices. The idea is to use the representational similarity matrices to characterize the representation space. CKA takes the activation matrices obtained from the network as inputs and gives a similarity score between 0 and 1 by comparing the similarities between all objects. Specifically, if $A_i \in \mathbb{R}^{L \times d_i}$ and $A_j \in \mathbb{R}^{L \times d_j}$ are the activation(representation) matrices for clients $i$ and $j$ for the RAD $\mathbf{X}$ of size $L$, the distance between $A_i$ and $A_j$ using Linear-CKA is computed as -

$$\text{Linear CKA}(A_i A_i^T, A_j A_j^T) = \text{Linear CKA}(K_i, K_j) = \frac{||A_j^T A_i||_F^2}{||A_i^T A_i||_F ||A_j^T A_j||_F}, \quad (3)$$

where $K_i$ and $K_j$ are kernel matrices for any choice of kernel $\mathcal{K}$ such that we have $K_i(p, q) = \mathcal{K}(A_i(p, :), A_i(q, :))$. CKA also allows choice of other kernels like RBF-kernel, polynomial kernel etc. but we observed that the results with Linear-CKA are as good as those with RBF kernel as also reported in (Kornblith et al., 2019). Thus we focus only on Linear-CKA here.

Finally, the local loss at each client becomes -

$$\min_{\mathcal{W}_k} \mathcal{L}_k = \ell(\mathcal{W}_k) + \mu \, d_{\text{Linear-CKA}}\Big(K_k, \bar{K}(t-1)\Big). \quad (4)$$

with $\bar{K}(t-1) = \sum_{j=1}^{N} w_j K_j(t-1)$ and $K_j(t-1) = \Phi_j(\mathbf{X}; \mathcal{W}_j(t-1)).\Phi_j(\mathbf{X}; \mathcal{W}_j(t-1))^T$. When $\mu = 0$, each client is training its own local models without any peer-supervision and when $\mu \to \infty$ the local models try to converge on the RAD representations without caring about the SSL loss.

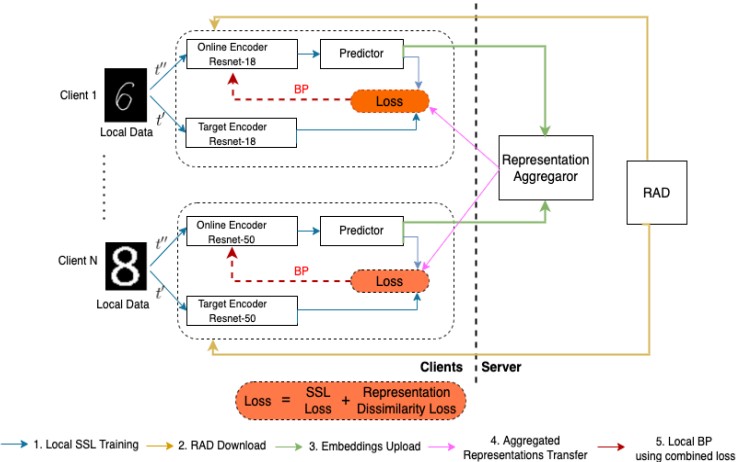

Figure 1: Overview of the proposed framework: Hetero-SSFL. In each training round, the server sends the RAD and the aggregated representation matrix to all clients, each client trains local self-supervised models by optimising the combined loss, the representations obtained using local model on RAD are sent back to the server which aggregates the representations and sends it back to all clients for next training round. Note: the local models are named Resnet-18 and Resnet-50 as examples to depict heterogeneous client models.

---

**Algorithm 1** Hetero-SSFL Algorithm for Heterogeneous clients

---

**Input:** number of clients $N$, number of global communication rounds $T$, number of local epochs $E$, parameter $\mu$, weight vector for clients $[w_1, w_2, \ldots w_N]$
**Output:** Final set of personalised models $\mathcal{W}_1(T), \mathcal{W}_2(T) \ldots \mathcal{W}_N(T)$
**At Server -**
Initialize $\mathcal{W}_1(0), \mathcal{W}_2(0) \ldots \mathcal{W}_N(0)$
**for** $t = 1$ **to** $T$ **do**
   RAD = $\mathbf{X}$
   **for** each client $j$ **do**
      $A_j(\mathbf{X}) = \Phi_j(\mathbf{X}; \mathcal{W}_j(t-1))$
      $K_j = \mathcal{K}(A_j(\mathbf{X}), A_j(\mathbf{X}))$
   **end for**
   $\bar{K}(t-1) = \sum_{j=1}^{N} w_j K_j$
   Select a subset of clients $\mathcal{N}_t$
   **for** each selected client $i \in \mathcal{N}_t$ **do**
      $\mathcal{W}_i(t) = \textbf{LocalTraining}(\mathcal{W}_i(t-1), \bar{K}(t-1), \mathbf{X}, \mu)$
   **end for**
**end for**
Return $\mathcal{W}_1(T), \mathcal{W}_2(T) \ldots \mathcal{W}_N(T)$
**LocalTraining**$(\mathcal{W}_i(t-1), \bar{K}(t-1), \mathbf{X}, \mu)$
Initialize $\mathcal{W}_i(t)$ with $(\mathcal{W}_i(t-1))$
**for** each local epoch **do**
   Generate local representation matrix $A_i(\mathbf{X}) = \Phi_i(\mathbf{X}; \mathcal{W}_i(t))$
   Generate local kernel matrix $K_i = \mathcal{K}(A_i(\mathbf{X}), A_i(\mathbf{X}))$
   Update $\mathcal{W}_i(t)$ after solving for loss in equation (4)
**end for**
Return $\mathcal{W}_i(t)$

---

## 4   CONVERGENCE ANALYSIS

In this section we provide insights on convergence of the proposed framework. The convergence guarantee is shown to hold under mild assumptions which are commonly made in the literature (Wang et al., 2020b; Tan et al., 2022).

**Assumption 4.1** (Lipschitz Smoothness). The local loss function on every client $k$, $\mathcal{L}_k$, is assumed to be $L_1$-Lipschitz smooth, which implies:

$$\mathcal{L}_k(a) - \mathcal{L}_k(b) \leq \nabla \mathcal{L}_k(b)^\top (\mathcal{W}_k(a) - \mathcal{W}_k(b)) + \frac{L_1}{2}||\mathcal{W}_k(a) - \mathcal{W}_k(b)||_2^2, \quad \forall \, a, b > 0. \quad (5)$$

**Assumption 4.2** (Stochastic Gradient). The local stochastic gradient for each client $k$, at any time $t$, $g_{k,t}$ is an unbiased estimator of the gradient, has bounded variance and the expectation of its norm is bounded above. That is, we have

$$\mathbb{E}[g_{k,t}] = \nabla \mathcal{L}_k \quad \text{and} \quad \text{Var}(g_{k,t}) \leq \sigma^2,$$
$$\mathbb{E}[||g_{k,t}||_2] \leq P.$$

**Assumption 4.3** (Representation Norm). The norm of the representations obtained for any instance on any one client is bounded above by $R$. That is:

$$||\Phi_k(x; \mathcal{W}_k)||_2 \leq R.$$

**Assumption 4.4** (Lipschitz Continuity). The embedding function $\Phi(.;.)$ for all clients is $L_2$-Lipschitz continuous, which for each client $k$ implies:

$$||\Phi_k(.; \mathcal{W}_a) - \Phi_k(.; \mathcal{W}_b)||_2 \leq L_2||\mathcal{W}_a - \mathcal{W}_b||_2.$$

We start with the following lemma on the reduction of the local loss after $E$ local epochs.

**Lemma 4.5.** *(Local Epochs) In each communication round, the local loss $\mathcal{L}$ for each client reduces after $E$ local epochs and is bounded as below under the Assumption 4.2.*

$$\mathbb{E}[\mathcal{L}_E] \leq \mathcal{L}_0 - (\eta - \frac{L_1\eta^2}{2}) \sum_{i=0}^{E-1} ||\nabla \mathcal{L}_i||^2 + \frac{L_1 E \eta^2}{2}\sigma^2. \quad (6)$$

*where $\mathcal{L}_0$ and $\mathcal{L}_E$ denote the loss before and after $E$ local epochs respectively and $\eta$ is learning rate.*

Proof of Lemma 4.5 is in Appendix A.1. Lemma 4.5, *similar to (Tan et al., 2022)*, shows the bound on the client's local loss after the completion of local epochs under one global communication round.

After the local training, the representations from all clients are sent to the server. Our next lemma 4.6 shows a bound on the expected loss after every global representation update at the server.

**Lemma 4.6.** *(Representation Update) In each global communication round, for each client, after the $E$ local updates, the global representation matrix is updated at the server and the loss function for any client $k$ gets modified to $\mathcal{L}_{E'}$ from $\mathcal{L}_E$ and could be bounded as -*

$$\mathbb{E}[\mathcal{L}_{E'}] \leq \mathbb{E}[\mathcal{L}_E] + 2\mu\eta L_2 P R^3 L^2. \quad (7)$$

Proof of Lemma 4.6 is in Appendix A.2.

Equipped with the results from Lemmas 4.5 and 4.6, we are now ready to state our main result which provides a total deviation bound for an entire training round. Theorem 4.7 shows the divergence in loss for any client after completion of one round. We can guarantee the convergence by appropriately choosing $\mu$ and $\eta$ which leads to a certain expected decrease in the loss function. The result holds for convex as well as non-convex loss functions.

**Theorem 4.7** (Convergence). *After one global round, the loss function of any client decreases and is bounded as shown below:*

$$\mathbb{E}[\mathcal{L}_{E'}] \leq \mathcal{L}_0 - (\eta - \frac{L_1\eta^2}{2}) \sum_{i=0}^{E-1} ||\nabla \mathcal{L}_i||^2 + \frac{L_1 E \eta^2}{2}\sigma^2 + 2\mu\eta L_2 P R^3 L^2. \quad (8)$$

*Thus, if we choose $\eta$ and $\mu$ such that*

$$\eta < \frac{2(\sum_{i=0}^{E-1} ||\nabla \mathcal{L}_i||^2 - 2\mu L_2 P R^3 L^2)}{L_1(\sum_{i=0}^{E-1} ||\nabla \mathcal{L}_i||^2 + E\sigma^2)}, \quad \mu < \frac{\sum_{i=0}^{E-1} ||\nabla \mathcal{L}_i||^2}{2L_2 P R^3 L^2},$$

*then the convergence of Algorithm 1 is guaranteed.*

The proof of Theorem 4.7 follows directly from the results of Lemmas 4.5 and 4.6.

## 5 EXPERIMENTS

In this section we describe the experimental setup and present results for our method alongside the popular baselines for the image recognition task as described above. The experiment for text based application is provided in Appendix B. We simulate system heterogeneity by randomly choosing different architectures for local client models. Within system heterogeneity we evaluate our model under varying statistical settings by manipulating the data distributions across clients to be IID or non-IID. We also demonstrate the performance of our framework with increase in number of clients. Some additional experimental results for this application are included in the Appendix C.

### 5.1 EXPERIMENTAL DETAILS

**Datasets** We test our framework on image classification task and use three datasets to compare against the baselines: CIFAR-10, CIFAR-100 and Tiny-Imagenet as suggested by several works in FL and the popular federated learning benchmark LEAF (Caldas et al., 2019). CIFAR-10 and CIFAR-100 contain 50,000 train and 10,000 test colored images for 10 classes and 100 classes respectively and Tiny-Imagenet has 100,000 images for 200 classes.

**Implementation Details** For the FL simulations, we explore two types of settings, non-IID setting and IID setting. In the IID-setting, we assume each client to have access to the data of all classes and in the non-IID setting we assume each client has access to data of only a few disjoint classes. For example, for CIFAR-10, client 1 might have access to objects of classes $\{1, 2\}$ versus client 2 with access to $\{3, 4\}$ and likewise. The IID setting is created by dividing all the instances between the clients. For the non-IID setting, for each client a fraction of classes is sampled and then instances are divided amongst the clients containing specific classes. In the cases when number of clients, $N$, is less than total number of classes $K$, each client is assumed to have access to all instances corresponding to $\frac{K}{N}$ classes. The choice of encoders at each client is data and application dependent. Since we are dealing with images, to create heterogeneous clients, for each client we randomly select between Resnet-18 and Resnet-34 models as encoders for our method. Since all the baselines work in homogeneous client settings and hence have to use the same model, we use Resnet-34 at each client in the baselines for comparison. The total number of global communication rounds is kept to 200 for all methods with each client performing $E = 5$ local epochs in each round. We use SGD with momentum $= 0.9$, batch size $= 200$ and learning rate $= .032$ for training all models. The weight vector used in aggregating the representations $\mathbf{w}$ is set to be uniform to $\frac{1}{n}$. All these models are trained on a machine with 4 GeForce RTX 3090 GPUs with each GPU having about 24gb of memory.

**Baselines** We compare our method against two very recently proposed federated self-supervised baselines: FedU and FedEMA, which reflect the current state-of-the-art. Since the source code for these baselines is not publicly available, we perform these comparisons using our implementation and with the best parameters reported in these papers. We also create additional baselines by combining central SSL algorithms like BYOL and SimSiam with the federated learning procedure of FedAvg to make FedBYOL and FedSimSiam. The performance of supervised FedAvg and centralised BYOL is also reported for comparisons as theoretical baselines. All of these methods however work in homogeneous settings, hence we compare the homogeneous setting of baselines with Resnet-34 architecture with our heterogeneous setting having a mix of Resnet-34 and Resnet-18 architectures. By doing this, we are comparing our framework with more powerful baselines and thus expect the improvement our baselines to be as good or better when doing an identical comparison. One additional baseline is the SSFL algorithm, but due to the unavailability of the source code and lack of precise implementation details in the paper, we could not compare against it.

**Evaluation** We follow the linear evaluation protocol as mentioned in (Zhuang et al., 2021) to test the models. Under this protocol, the local encoders are trained in self-supervised way without any labeled data and then a linear classifier is trained on top of the encoder after freezing the encoder parameters. We use a linear fully connected layer with 1000 neurons as the classifier and train it for 100 epochs with Adam optimizer, batch size = 512 and learning rate = 0.003. The reported results are the accuracy of the linear classifier on a separate test dataset.

**Other parameters** The other parameters important for our approach are $\mu$ which controls the weight of the proximal term in the loss function, and the size of the RAD dataset. We tune the hyperparameter $\mu$ using a validation set and found the best values of $\mu$ to be $0.1$ for the CIFAR datasets and $1$ for the Tiny-Imagenet dataset. For the RAD size we observe that increasing the size of RAD increases performance but has an effect on the speed of simulation so we fix that size to be

Table 1: Test Accuracy comparison of our method with baselines under the linear evaluation protocol in non-IID settings.

| Method | CIFAR-10 | CIFAR-100 | Tiny-Imagenet |
|---|---|---|---|
| FedBYoL | $78.1 \pm 1.1$ | $58.1 \pm 0.3$ | $31.8 \pm 2.0$ |
| FedSimSiam | $76.7 \pm 1.5$ | $53.2 \pm 0.16$ | $29.1 \pm 1.2$ |
| FedU | $79.6 \pm 0.5$ | $58.9 \pm 0.12$ | $58.23 \pm 2.3$ |
| FedEMA | $81.2 \pm 1.6$ | $61.8 \pm 0.31$ | $58.2 \pm 3.1$ |
| Hetero-SSFL | $\mathbf{90.29 \pm 0.3}$ | $\mathbf{66.1 \pm 0.21}$ | $\mathbf{61.5 \pm 1.8}$ |
| Supervised FedAvg (Supervised upper bound) | $68.5 \pm 1.7$ | $44.29 \pm 1.34$ | $27.2 \pm 3.2$ |
| Central-BYOL (IID upper bound) | $94.3 \pm 0.2$ | $74.2 \pm 0.67$ | $78.7 \pm 1.3$ |

5000 in our experiments. On studying the effect of varying number of local epochs we observed that unlike the FedAvg based algorithms our method is stable with increasing local epochs.

Table 2: Test Accuracy comparison of our method with baselines under the linear evaluation protocol in IID settings(with equal number of classes on all clients).

| Dataset | FedBYoL | FedSimSiam | FedU | FedEMA | Hetero-SSFL |
|---|---|---|---|---|---|
| CIFAR-10 | $82.24 \pm 0.8$ | $80.4 \pm 1.2$ | $81.6 \pm 1.8$ | $85.9 \pm 0.3$ | $\mathbf{88.5 \pm 1.3}$ |
| CIFAR-100 | $23.8 \pm 0.14$ | $22.19 \pm 1.8$ | $20.4 \pm 0.91$ | $25.7 \pm 0.4$ | $\mathbf{34.8 \pm 0.7}$ |

## 5.2 RESULTS

We show the performance of our method in comparison with baselines in non-IID settings in Table 1 and that for IID settings in Table 2. We observe that our method outperforms the baselines by a significant margin and is slightly worse than the central BYOL (requires the data to be present on the central server and is not federated) which implies that we are able to achieve successful collaboration amongst clients. We also report the performance of our method and high-performing baselines with increase in scale in Table 3. As the number of clients increase, in every round only a fraction of clients are sampled to perform local training, total 20 clients with 5 clients selected per round is denoted as 20clients(5) in Table 3.

Table 3: Test Accuracy of various federated self-supervised methods with change in scale on CIFAR-100 dataset for the non-IID setting.

| Dataset | 5clients(5) | 20clients(5) | 100clients(10) |
|---|---|---|---|
| FedU | $58.9 \pm 0.12$ | $48.29 \pm 1.8$ | $41.1 \pm 2.3$ |
| FedEMA | $61.8 \pm 0.31$ | $52.5 \pm 0.8$ | $42.0 \pm 2.7$ |
| Hetero-SSFL | $66.1 \pm 0.21$ | $57.5 \pm 0.9$ | $43.3 \pm 1.9$ |
| FedAvg (Supervised Upper Bound) | $65.6 \pm 0.5$ | $60.68 \pm 0.23$ | $55.5 \pm 1.74$ |

## 6 CONCLUSION

In this work we study the problem of self-supervised federated learning with heterogeneous clients and propose a novel framework to enable collaboration in such settings. The framework allows clients with varying data/compute resources to partake in the collaborative training procedure. This development enables several practical settings to still gainfully employ self-supervised FL even though the different clients, for example different edge-devices or distinct organisations, might not have similar compute and data resources. The high performance of the framework across multiple datasets indicates its promise in achieving collaboration while the theoretical results guarantee the convergence of the algorithm for fairly general loss functions. In future work we plan to investigate the use of attention based mechanisms for adaptive weighting of different clients to generate aggregate representations based on each client's relative importance. Also, rather than assuming that the server has access to an unlabeled dataset and transmitting RAD from the server to clients, we can create a common generative model in a federated way like federated GANs to generate data instances required for RAD.

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

# Supplement for "Federated Self-supervised Learning for Heterogeneous Clients"

In this supplementary material, we provide proofs for the key results in the paper in Appendix A along with the intuitive explanations of the results. Then, we present a setting to illustrate the framework's utility in the text classification setting in Appendix B, and provide additional experiments in Appendix C.

## A   PROOFS

We first provide proof of Lemma 4.5 in Appendix A.1. Then, the proof of Lemma 4.6 is given in Appendix A.2.

### A.1   PROOF OF LEMMA 4.5

Let $\mathcal{L}_E$ denote the loss function after $E$ local epochs. Then, from the Lipschitz smoothness assumption in Assumption 4.1, we obtain that

$$\mathcal{L}_1 \leq \mathcal{L}_0 - \eta[\nabla \mathcal{L}_0^T g_0] + \frac{L_1 \eta^2}{2}[||g_0||^2].$$

Therefore, we have

$$\mathbb{E}[\mathcal{L}_1] \leq \mathbb{E}[\mathcal{L}_0] - \eta \mathbb{E}[\nabla \mathcal{L}_0^T g_0] + \frac{L_1 \eta^2}{2}\mathbb{E}[||g_0||^2]$$

$$= \mathbb{E}[\mathcal{L}_0] - \eta||\nabla \mathcal{L}_0||^2 + \frac{L_1 \eta^2}{2}\mathbb{E}[||g_0||^2]$$

$$= \mathbb{E}[\mathcal{L}_0] - \eta||\nabla \mathcal{L}_0||^2 + \frac{L_1 \eta^2}{2}(Var(g_0) + \mathbb{E}[||g_0||]^2)$$

$$\leq \mathbb{E}[\mathcal{L}_0] - (\eta - \frac{L_1 \eta^2}{2})||\nabla \mathcal{L}_0||^2 + \frac{L_1 \eta^2}{2}\sigma^2.$$

By summing the above bounds over $E$ number of local epochs, we find that

$$\sum_{i=1}^{E} \mathbb{E}[\mathcal{L}_i] \leq \sum_{i=0}^{E-1} \mathbb{E}[\mathcal{L}_i] - (\eta - \frac{L_1 \eta^2}{2}) \sum_{i=0}^{E-1} ||\nabla \mathcal{L}_i||^2 + \frac{L_1 E \eta^2}{2}\sigma^2$$

$$\mathbb{E}[\mathcal{L}_E] \leq \mathcal{L}_0 - (\eta - \frac{L_1 \eta^2}{2}) \sum_{i=0}^{E-1} ||\nabla \mathcal{L}_i||^2 + \frac{L_1 E \eta^2}{2}\sigma^2.$$

Therefore, if $\eta < \dfrac{2(\sum_{i=0}^{E-1} ||\nabla \mathcal{L}_i||^2)}{L_1(\sum_{i=0}^{E-1} ||\nabla \mathcal{L}_i||^2 + E\sigma^2)}$, we obtain $\mathcal{L}_E \leq \mathcal{L}_0$ and the local loss reduces after $E$ local epochs. As a consequence, we obtain the conclusion of the lemma.

### A.2   PROOF OF LEMMA 4.6

For any arbitrary client after the global representation update step, if the loss function gets modified to $\mathcal{L}_{E'}$ from $\mathcal{L}_E$, we have

$$\mathcal{L}_{E'} = \mathcal{L}_E + \mathcal{L}_{E'} - \mathcal{L}_E,$$

$$= \mathcal{L}_E + \mu \Big( d(.;t) - d(.;t-1) \Big).$$

As for client $i$, $d(.;t) = \text{Linear-CKA}(K_i(t), \bar{K}(t))$, we have,

$$\mathcal{L}_{E'} = \mathcal{L}_E + \mu \Big( \text{Linear-CKA}(K_i(t), \bar{K}(t)) - \text{Linear-CKA}(K_i(t), \bar{K}(t-1)) \Big).$$

Since this holds for all clients, we drop the client index $i$ going forward and obtain that

$$
\begin{aligned}
\mathcal{L}_{E'} &= \mathcal{L}_E + \mu\Big(\text{trace}(K(t)\bar{K}(t)) - \text{trace}(K(t)\bar{K}(t-1))\Big) \\
&= \mathcal{L}_E + \mu\Big(\text{trace}(K(t)(\bar{K}(t) - \bar{K}(t-1)))\Big) \\
&= \mathcal{L}_E + \mu\Big(\text{trace}(K(t)(\sum_{k=1}^{N} w_k K_k(t) - \sum_{k=1}^{N} w_k K_k(t-1)))\Big) \\
&= \mathcal{L}_E + \mu\Big(\sum_{i=1}^{L}\sum_{j=1}^{L}(K(t)_{i,j}(\sum_{k=1}^{N} w_k K_k(t)_{i,j} - \sum_{k=1}^{N} w_k K_k(t-1)_{i,j})\Big) \\
&= \mathcal{L}_E + \mu\Big(\sum_{k=1}^{N} w_k(\sum_{i=1}^{L}\sum_{j=1}^{L} K(t)_{i,j}(K_k(t)_{i,j} - K_k(t-1)_{i,j}))\Big) \\
&= \mathcal{L}_E + \mu\Big(\sum_{k=1}^{N} w_k(\sum_{i=1}^{L}\sum_{j=1}^{L} K(t)_{i,j}(\Phi_k(x_i;\mathcal{W}_k^t).\Phi_k(x_j;\mathcal{W}_k^t) - \Phi_k(x_i;\mathcal{W}_k^{t-1}).\Phi_k(x_j;\mathcal{W}_k^{t-1})))\Big).
\end{aligned}
$$

Given the above equations, we find that

$$
\begin{aligned}
\mathcal{L}_{E'} = \mathcal{L}_E + \mu\sum_{k=1}^{N} w_k \sum_{i=1}^{L}\sum_{j=1}^{L} K(t)_{i,j}\Big(&\Phi_k(x_i;\mathcal{W}_k^t)(\Phi_k(x_j;\mathcal{W}_k^t) - \Phi_k(x_j;\mathcal{W}_k^{t-1})) \\
&+ \Phi_k(x_j;\mathcal{W}_k^{t-1})(\Phi_k(x_i;\mathcal{W}_k^t) - \Phi_k(x_i;\mathcal{W}_k^{t-1})).\Big)
\end{aligned}
$$

Taking norm on both the sides, we get

$$
\begin{aligned}
||\mathcal{L}_{E'}|| = \Big\|&\mathcal{L}_E + \mu\sum_{k=1}^{N} w_k \sum_{i=1}^{L}\sum_{j=1}^{L} K(t)_{i,j}\Big(\Phi_k(x_i;\mathcal{W}_k^t)(\Phi_k(x_j;\mathcal{W}_k^t) - \Phi_k(x_j;\mathcal{W}_k^{t-1})) \\
&+ \Phi_k(x_j;\mathcal{W}_k^{t-1})(\Phi_k(x_i;\mathcal{W}_k^t) - \Phi_k(x_i;\mathcal{W}_k^{t-1}))\Big)\Big\| \\
\leq ||\mathcal{L}_E|| + \mu\Big\|&\sum_{k=1}^{N} w_k \sum_{i=1}^{L}\sum_{j=1}^{L} K(t)_{i,j}\Big(\Phi_k(x_i;\mathcal{W}_k^t)(\Phi_k(x_j;\mathcal{W}_k^t) - \Phi_k(x_j;\mathcal{W}_k^{t-1})) \\
&+ \Phi_k(x_j;\mathcal{W}_k^{t-1})(\Phi_k(x_i;\mathcal{W}_k^t) - \Phi_k(x_i;\mathcal{W}_k^{t-1}))\Big)\Big\| \\
\leq \mathcal{L}_E + \mu&\sum_{k=1}^{N} w_k \sum_{i=1}^{L}\sum_{j=1}^{L} ||K(t)_{i,j}||\Big(||\Phi_k(x_i;\mathcal{W}_k^t)(\Phi_k(x_j;\mathcal{W}_k^t) - \Phi_k(x_j;\mathcal{W}_k^{t-1})) \\
&+ \Phi_k(x_j;\mathcal{W}_k^{t-1})(\Phi_k(x_i;\mathcal{W}_k^t) - \Phi_k(x_i;\mathcal{W}_k^{t-1}))||\Big) \\
\leq \mathcal{L}_E + \mu&\sum_{k=1}^{N} w_k \sum_{i=1}^{L}\sum_{j=1}^{L} ||K(t)_{i,j}||\Big(||\Phi_k(x_i;\mathcal{W}_k^t)||.||(\Phi_k(x_j;\mathcal{W}_k^t) - \Phi_k(x_j;\mathcal{W}_k^{t-1}))|| \\
&+ ||\Phi_k(x_j;\mathcal{W}_k^{t-1})||.||(\Phi_k(x_i;\mathcal{W}_k^t) - \Phi_k(x_i;\mathcal{W}_k^{t-1}))||\Big) \\
\leq \mathcal{L}_E + \mu&\sum_{k=1}^{N} w_k \sum_{i=1}^{L}\sum_{j=1}^{L} ||K(t)_{i,j}||\Big(||\Phi_k(x_i;\mathcal{W}_k^t)||.||\text{L}_2\eta g_{t,k}|| + ||\Phi_k(x_j;\mathcal{W}_k^{t-1})||.||\text{L}_2\eta g_{t,k}||\Big).
\end{aligned}
$$

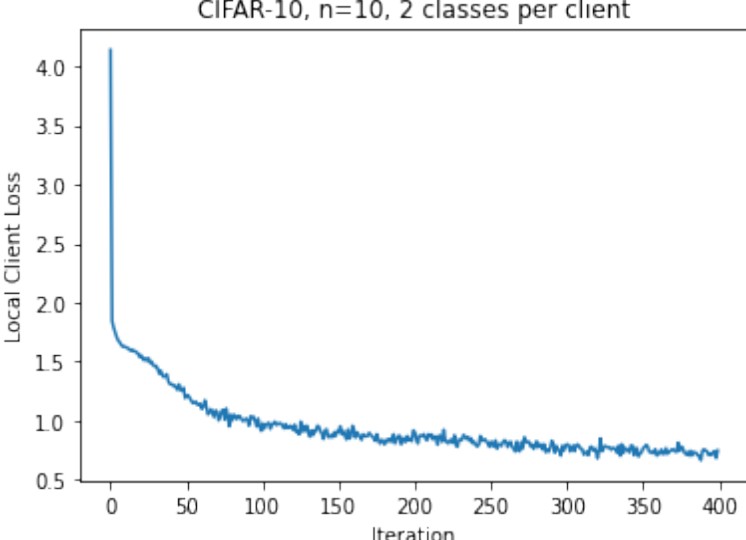

Figure 2: Local loss at the end of each training round for a randomly selected client.

Taking expectation on both sides of the above bounds, we have

$$\mathbb{E}[\mathcal{L}_{E'}] \leq \mathbb{E}[\mathcal{L}_E] + \mu \sum_{k=1}^{N} w_k \sum_{i=1}^{L} \sum_{j=1}^{L} ||K(t)_{i,j}|| \left( \mathbb{E}[||\Phi_k(x_i; \mathcal{W}_k^t)||.||\mathrm{L}_2 \eta g_{t,k}||] + \mathbb{E}[||\Phi_k(x_j; \mathcal{W}_k^{t-1})||.||\mathrm{L}_2 \eta g_{t,k}||] \right)$$

$$\leq \mathbb{E}[\mathcal{L}_E] + \mu \sum_{k=1}^{N} w_k \sum_{i=1}^{L} \sum_{j=1}^{L} ||K(t)_{i,j}|| \left( R\mathrm{L}_2 \eta P + R\mathrm{L}_2 \eta P \right)$$

$$= \mathbb{E}[\mathcal{L}_E] + 2\mu\eta\mathrm{L}_2 PR \sum_{i=1}^{L} \sum_{j=1}^{L} ||K(t)_{i,j}||$$

$$\leq \mathbb{E}[\mathcal{L}_E] + 2\mu\eta\mathrm{L}_2 PR \sum_{i=1}^{L} \sum_{j=1}^{L} R^2$$

$$\leq \mathbb{E}[\mathcal{L}_E] + 2\mu\eta\mathrm{L}_2 PR^3 L^2.$$

Thus, we have,

$$\mathbb{E}[\mathcal{L}_{E'}] \leq \mathbb{E}[\mathcal{L}_E] + 2\mu\eta\mathrm{L}_2 PR^3 L^2,$$

which concludes the proof.

### A.3 CONVERGENCE DEMONSTRATION IN EXPERIMENTS

We monitor the local loss at the end of each training round and we see that the decreasing loss theorem holds in practice as well. The Figure 2 shows a plot of the local loss for one randomly selected client with the number of global round for the proposed method on CIFAR-10 dataset.

## B APPLICATIONS IN OTHER DOMAINS

While the experiments with image recognition tasks demonstrated the potential of the approach in popular federated self-supervised settings, note that the Hetero-SSFL framework is indeed extensible to other domains and applications as well since the key idea of the framework (i.e., enhance local self-supervised learning on heterogeneous clients with limited data by obtaining peer supervision from other clients) is broadly applicable. We now briefly illustrate how the proposed method can be applied for the text classification task.

Consider each client to be learning a transformer for language modeling using the self-supervised Masked Language Modeling(MLM) objective. This objective tries to reconstruct the sentence by predicting the masked words in the sentence. Thus, the local self-supervised loss function for client $k$, $\ell(\mathcal{W}_k)$, corresponds to the MLM loss. The peer supervision is achieved as usual, by server obtaining the embeddings on the RAD from each client, aggregating those embeddings and sending it to the clients for the next round of training. The local loss function at each client is similar to Equation (4) but with the new $\ell(\mathcal{W}_k)$. After pre-training using self-supervision, we evaluate the performance of training by following a linear evaluation protocol using a sentiment classification task on Sentiment140 dataset from LEAF (Caldas et al., 2019).

**Experimental Setup** For local self-supervised learning we use ALBERT transformer (Lan et al., 2020). We partition the Sentiment140 data in 5 non-iid partitions for each client and fine-tune the transformer for each client on locally available data for 200 rounds, using AdamW optimizer and a range of $\mu$ values between .01 and 100 and report the best performance. After pre-training, the parameters of the transformer are frozen and a linear classifier is trained on the sentence embeddings obtained from the transformer. The test accuracy of the linear classifier is reported as the result.

For baselines, we consider the client's local model training on local data, FedAvg version of ALBERT transformer and a divergence aware update method as shown in the baselines FEDEMA and FedU. Since the other self-supervised federated methods like FedEMA and FedU consider only the image classification task, we are unable to compare against them.

The Table 4 below shows the test accuracy of Hetero-SSFL in comparison with the above mentioned baselines.

Table 4: Test accuracy for different methods for text classification on Sentiment140 dataset.

| | Method | | | |
| --- | --- | --- | --- | --- |
| | FedAvg | Local | Divergence-Aware | Hetero-SSFL |
| Test Accuracy | $61.53 \pm 7.2$ | $61.7 \pm 2.2$ | $58.19 \pm 9.2$ | $\mathbf{62.79 \pm 1.7}$ |

## C  ADDITIONAL EXPERIMENTS

Table 5: Top-1 test accuracy for different methods under semi-supervised learning for CIFAR100.

| Method | 1% labeled data | 10% labeled data |
| --- | --- | --- |
| FedU | $20.5 \pm 0.5$ | $46.3 \pm 0.57$ |
| FedEMA | $23.11 \pm 0.65$ | $48.6 \pm 0.47$ |
| Hetero-SSFL | $\mathbf{37.4 \pm 0.95}$ | $\mathbf{57.4 \pm 0.43}$ |

Table 6: Top-1 test accuracy for different methods under semi-supervised learning for CIFAR10.

| Method | 1% labeled data | 10% labeled data |
| --- | --- | --- |
| FedU | $72.9 \pm 2.25$ | $86.8 \pm 0.5$ |
| FedEMA | $75.8 \pm 1.4$ | $87.4 \pm 0.8$ |
| Hetero-SSFL | $\mathbf{79.3 \pm 1.85}$ | $\mathbf{88.7 \pm 0.25}$ |

Table 7: Average effect of collaboration on different clients. Column 2 depicts the test accuracy of clients with local models and column 3 the accuracy under Hetero-SSFL.

| Client Model Architecture | Local SSL | Hetero-SSFL |
| --- | --- | --- |
| Resnet-18 | 71.9 | $\mathbf{85.4}$ |
| Resnet-34 | 77.8 | $\mathbf{93.7}$ |

We also test our framework on the semi-supervised learning task, as described in FedU (Zhuang et al., 2021) and FedEMA (Zhuang et al., 2022) papers. Under this protocol, we let the federated SSFL

methods train the local models for representation learning. For the evaluation, we assume that only a small fraction($10\%$ or $1\%$) of labeled data is available. Then, after the federated representation learning of local models on clients, we fine-tune the local models along with a linear classification layer on top for 100 epochs. We evaluate our method and the baselines, FedU and FedEMA, in this setting on CIFAR10 and CIFAR100 datasets and observe that our method outperforms the baselines. For fine-tuning the models, we use SGD with Nesterov momentum as the optimizer with momentum = 0.9, learning rate = 0.05 and batch size = 512 and 256 for $10\%$ labeled data and $1\%$ labeled data respectively. The results reported in Table 5 and Table 6 are for federated setting with non-IID clients with 5 users, 20 classes per user for CIFAR100 and 2 classes per user for CIFAR10.

For additional insights, we show the average change in test accuracy of clients with different architectures when performing standalone training versus collaborative learning in the heterogeneous setting in Table 7.

# D    DETAILS ABOUT RAD

The Representation Alignment Dataset (RAD) is important for training in Hetero-SSFL. Functionally, the RAD is used to guide the simultaneous training on the peer clients by serving as a common dataset for representation alignment. Also, since the models are not transmitted and only the RAD representations are transmitted from clients to the server, there is no privacy loss due to involvement of RAD in the training process. The only assumption about this dataset is to be of the same domain as the training data, i.e., for image classification tasks it could be images sourced from the web and for the text-classification tasks it could be publicly available text like from Wikipedia, Reddit etc. Because the only requirement for RAD is to come from the same domain, it does not pose practical constraints. Several applications can make use of publicly available datasets to use for training in Hetero-SSFL, for example, learning self-supervised models for image datasets could use MNIST or CIFAR datasets for training, similarly acoustic applications can use audio datasets provided by Huggingface, and likewise.

In the experiments described in Table 1 and Table 2, we subset a part of the dataset (5000 data points) and set it aside to be used as RAD before beginning the training process. Since the RAD is considered generic with no bearing on clients, it is created before the clients are intialised and data is partitioned into clients. We further analysed the effect of using simple MNIST dataset as an RAD and saw similar results. The change in performance with respect to the size of RAD is given below in the Figure 3. When the RAD size is very low, the local models performance is closer to the performance in standalone training. As the size of the RAD grows, the performance gets better but because the gain is small after 3000 points, we keep $L = 3000$ for all the experiment simulations. Also, in cases when $\mu = 0$, the performance of the Hetero-SSFL is same as standalone training ensuring that the performance gain is not due to additional data in the form of RAD.

# E    COMMUNICATION COST ANALYSIS

In Hetero-SSFL, each global round involves the server sending the common dataset, RAD, to each client and the clients sending back the representations obtained on the RAD. The size of the RAD is given by $L$, thus the communication cost in each round is $O(L)$. In all our experiments, we set the RAD size between 3000 and 5000. Thus the communication cost in each round of Hetero-SSFL corresponds to transferring these many (3000-5000) parameters twice. This is orders of magnitude less than the conventional FL algorithms that require sharing of the model parameters which run in millions. For federated training of the Resnet18 models, having 11M number of parameters, Hetero-SSFL with $L = 5000$ will require 2000 times less communication cost than conventional methods in each round, making it very practical.

# F    ABLATION STUDIES

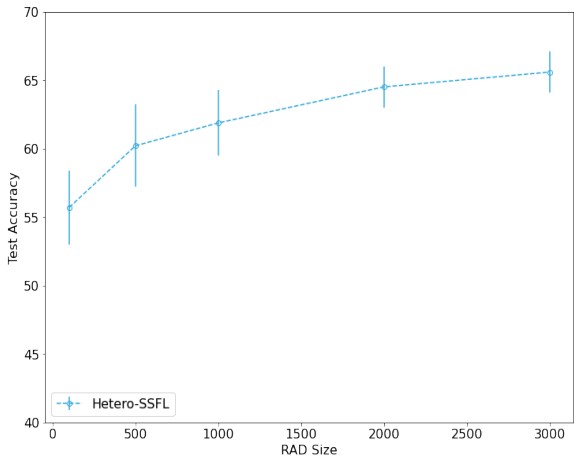

Figure 3: Change in performance with change in RAD size for CIFAR-100 dataset in non-IID setting.

Table 8: Test Accuracy comparison of our method with increasing heterogeneity for CIFAR-100 datasets and non-IID setting.

| Setting | Small Models | Resnet Models |
|---|---|---|
| 20 % small | $39.4 \pm 1.55$ | $64.2 \pm 2.9$ |
| 40 % small | $37.5 \pm 1.96$ | $63.7 \pm 1.89$ |
| Standalone Training | $28.8 \pm 0.9$ | $55.3 \pm 1.3$ |

### F.1 Increased Heterogeneity in Clients

To demonstrate the effectiveness of the framework with diverse types of heterogeneous clients, we experiment by adding clients with models of much smaller capacities like CNN and MLP along with Resnet18 and Resnet34 models. We use Hetero-SSFL in this setup on CIFAR-100 dataset to achieve federated training of self-supervised models on CIFAR-100 dataset and use linear evaluation protocol, as described in Section 5.1 to measure the test accuracy. We show the average performance of the small capacity clients, and average performance of the large capacity clients in settings with $20\%$ and $40\%$ small capacity models in Table 8 as compared to the standalone training.

### F.2 Effect of $\mu$

The parameter $\mu$ regulates the weight of the proximal term (used for peer alignment) against the local self-supervised loss, as shown in Equation 1. Thus, the value of $\mu$ plays an important role in the entire training process. $\mu = 0$ corresponds to standalone training where each client is minimizing the local self-supervised loss on its own without any federation. As the $\mu$ increases to $0.001, 0.01$, some federation starts to happen and the performance gets better than the standalone training. $\mu = 0.1$ achieves a good balance of the two loss functions for CIFAR datasets, and $\mu = 1$ for Tiny-Imagenet. But as the $\mu$ starts increasing beyond 1, the weight of the proximal term becomes higher than the self-supervised loss leading to a drop in performance. This effect is more pronounced when the heterogeneous models are diverse, setting $\mu = 10$ in those settings inhibits learning at the more powerful clients while still improving the performance on the less powerful clients. With this knowledge, we can potentially vary $\mu$ across clients depending on their local resources, for example, a client with less number of data points and lower training capacity could benefit more from proximal term by setting a higher $\mu$.

