# OpenReview forum: "Federated Self-supervised Learning for Heterogeneous Clients"
_ICLR.cc/2023/Conference — Submitted to ICLR 2023_

### Official Review · Reviewer_X5at · 2022-10-19

**Confidence:** 3
**Correctness:** 3
**Technical Novelty And Significance:** 2
**Empirical Novelty And Significance:** 3
**Recommendation:** 5

**Clarity, Quality, Novelty And Reproducibility:**

Overall, the paper is easy to follow but as mentioned in weaknesses, some details were missing, which might cause confusion to the audience. The technical contributions of the paper were quite marginal despite its superior performance compared with baseline methods. As for reproducibility, it is only possible when the RAD dataset is available.

**Details Of Ethics Concerns:**

There is a lack of discussion about privacy in federated learning and the authors should talk about potential privacy problems and whether the method can preserve differential privacy.

**Strength And Weaknesses:**

Strengths:
- The modification by adding an objective to make the representation on a common dataset aligned is intuitive and clear.
- Authors gave a convergence guarantee for the proposed method, making the approach more thorough and convincing from the theoretical side.

Weaknesses:
- Experiments were not very comprehensive. Some important ablation studies were missing. For example, how will the coefficient $\mu$ will affect the performance? And what is the influence of the size of the common dataset RAD? It is not clear whether the performance gain was due to the additional data.
- It was not clear how the server utilized the representations from different local clients, given that the dimensions were not the same. In Eq. (1), the second term was the distance of two representations, while in Eq. (4), it was changed to two kernels. It seems that the paper did not mention how to aggregate representations.
- Comparison of communication efficiency in federated learning is necessary in terms of metrics such as the size of transmitted data in one round.
- Some implementation details were not complete. For example, how to obtain the RAD dataset? The authors also stated that weight vector $[w_1, w_2, \dots, w_N]$ can be learned, but how to achieve it? Will learning the weight vector require the communication between the server and clients just as FedAvg to average weights from different clients?
- There is a lack of discussion about privacy in federated learning and the authors should talk about potential privacy problems and whether the method can preserve differential privacy.



**Summary Of The Paper:**

The paper proposed a new approach called Hetero-SSFL, to conduct federated learning under self-supervised tasks with heterogeneous clients. Specifically, besides local training, all clients learn to align the lower dimensional representations on a common dataset. The paper provides a convergence guarantee theoretically and demonstrate the improved performance empirically.

**Summary Of The Review:**

The paper has some important details missing and the experimental part was not comprehensive enough. The current version is not ready for publication.

---

> ### Author Response · Authors · 2022-11-17
> **Response to Reviewer X5at**
>
> We thank the reviewer for the useful feedback. We address the concerns raised by the reviewer below -
>
> 1. We have added a section, Section F Ablation Studies, in the Appendix for providing details about the common public dataset aka RAD. A subset of the section reads "The parameter $\mu$ regulates the weight of the proximal term (used for peer alignment) against the local self-supervised loss, as shown in Equation 1 of the paper. Thus, the value of $\mu$ plays an important role in the entire training process. $\mu = 0$ corresponds to standalone training where each client is minimizing the local self-supervised loss on its own without any federation. As the $\mu$ increases to $0.001, 0.01$, some federation starts to happen and the performance gets better than the standalone training. $\mu = 0.1$ achieves a good balance of the two loss functions for CIFAR datasets, and $\mu = 1$ for Tiny-Imagenet. But as the $\mu$ starts increasing beyond 1, the weight of the proximal term becomes higher than the self-supervised loss leading to a drop in performance. This effect is more pronounced when the heterogeneous models are diverse, setting $\mu = 10$ in those settings inhibits learning at the more powerful clients while still improving the performance on the less powerful clients. With this knowledge, we can potentially vary $\mu$ across clients depending on their local resources, for example, a client with less number of data points and lower training capacity could benefit more from proximal term by setting a higher $\mu$."
>
> 2. We have added a section, Section D Details About RAD, in the Appendix for providing details about the common public dataset aka RAD. A subset of the section reads ''The change in performance with respect to the size of RAD is given in the attached figure. When the RAD size is very low, the local models overfit on the RAD and do not learn robust representations which are evaluated for classification task. As the size of the RAD grows, the performance gets better but because the gain is small after 3000 points, we keep $L = 3000$ for all the experiment simulations. For Hetero-SSFL too, the RAD is used just used for aligning the representations across clients. In cases when $\mu=0$, the performance of the Hetero-SSFL is same as standalone training ensuring that the performance gain is not due to additional data in the form of RAD." Link to the figure - https://imgur.com/Q5bNBhk
>
> 3. We modified the paper to include a formula for the kernel computation using the representations.
>
> 4. We added a section in the Appendix regarding the communication cost analysis. A subset of the section reads - "In Hetero-SSFL, each global round involves the server sending the common dataset, RAD, to each client and the clients sending back the representations obtained on the RAD. The size of the RAD is given by $L$, thus the communication cost in each round is $O(L)$. In all our experiments, we set the RAD size to $3000$. Thus the communication cost in each round of Hetero-SSFL corresponds to transferring $3000$ parameters twice. This is orders of magnitude less than the conventional FL algorithms that require sharing of the model parameters which run in millions. For federated training of the Resnet18 models, having 11M number of parameters, Hetero-SSFL with $L = 5000$ will require 2000 times less communication cost in each round making it very practical."
>
> 5. We have added a section, Section D Details About RAD, in the Appendix for providing details about the common public dataset aka RAD. A subset of the section reads "The Representation Alignment Dataset (RAD) is important for training in Hetero-SSFL. Functionally, the RAD is used to guide the simultaneous training on the peer clients by serving as a common dataset for representation alignment. The only assumption about this dataset is to be of the same domain as the training data, i.e., for image classification tasks it could be images sourced from the web and for the text-classification tasks it could be publicly available text like from Wikipedia, Reddit etc. In the experiments described in Table 1 and Table 2 of the paper, we subset a part of the dataset (5000 data points) and set it aside to be used as RAD before beginning the training process. This dataset is created before the clients are intialised and data is partitioned into clients. We further analysed the effect of using simple MNIST dataset as an RAD and saw similar results."
>
> [continued in the next comment]

---

> > ### Author Response · Authors · 2022-11-17
> > **Response to Reviewer X5at**
> >
> > 6. The use of the weight vector, $[w_1, \dots w_N]$ is for aggregating the individual client representations to create the global representation matrix $\bar{\Phi}$, where each client $i$'s representation is given a weight of $w_i$. The learning of the weight vector at local client could be thought of as attention weights to be given to the representations (embeddings) of all other clients. The learning of the weight vector is different from federated learning and involves local learning at client to identify how much importance to give each client for aligning the representations.
> >
> > 7. Since the data from the client is not used in any way (raw data, transformed data, model weights learned using the local data) outside the client, the framework does not violate the local client privacy. The RAD in itself is assumed to be a public common dataset and only its' representations are shared outside of the client. Proving the differential privacy guarantee of the framework will be taken up in the future work. We thank the reviewer for suggesting it.

---

> > > ### Author Response · Authors · 2022-11-21
> > > **Regarding our response**
> > >
> > > Dear Reviewer X5at,
> > >
> > > We have addressed your concerns in our responses. We would like to hear your feedback. Please feel free to raise questions if you have other concerns.
> > >
> > > Best regards,
> > >
> > > Authors

---

> > > ### Author Response · Authors · 2022-11-29
> > > **Look forward to your feedback.**
> > >
> > > Dear Reviewer X5at,
> > >
> > > We have addressed your concerns in our responses. We would like to hear your feedback. Please feel free to raise questions if you have other concerns.
> > >
> > >
> > > Thanks and regards,
> > >
> > > Authors

---

> > ### Comment · Reviewer_X5at · 2022-12-03
> > **Post-rebuttal comments**
> >
> > Thanks for your response. I appreciate that some missing details were added and discussed in the paper such as the hyperparameter study and the choice of RAD dataset. However, for the communication cost analysis, I think the size L is the number of images to be transmitted rather than the real size (the number of float parameters). For a fair comparison, it is necessary to consider the size of the image (how many pixels in an image) as well. And still, I feel that the technical contributions were marginal given the current form of the paper.

---

> > > ### Author Response · Authors · 2022-12-05
> > > **Response to Reviewer X5at**
> > >
> > > We thank the reviewer for the response.
> > >
> > > We would also like to clarify that L is the number of images used in RAD and because the RAD images are public, it is not necessary to transfer images from server to clients. The metadata regarding the images like source, indices is sufficient for the clients to identify the images. Moreover, even when the transfer becomes must, the communication cost of sharing images is a one time cost that is incurred at the beginning of training process, as opposed to a fixed cost per iteration which is incurred in sharing models.
> > >
> > > Best,
> > >
> > > Authors

---

### Official Review · Reviewer_N6gv · 2022-10-24

**Confidence:** 4
**Clarity, Quality, Novelty And Reproducibility:** This paper is well written and addres…
**Correctness:** 3
**Technical Novelty And Significance:** 3
**Empirical Novelty And Significance:** 3
**Recommendation:** 6

**Strength And Weaknesses:**

### Strengths

*    The paper is fairly well written, although it would benefit if Figure 1 was presented earlier in the text (maybe at the top of page 4?) Making it more self-contained would be good too.

*    The results in Tables 1,2,3 look super good compared to the other baselines but. Could the Authors explain why the figures in Table 2 are lower than in Table 1? It seems like all the baselines do better in the IID case (as you'd normally expect) for CIFAR-10 but Hetero-SSFL does worse... But then for CIFAR-100 in Table 2 (IID) all methods do worse than in Table 1 (non-IID). Could the Authors elaborate? Maybe the numbers got mixed up adding them to the table?

### Weaknesses
Some weaknesses and clarifications needed:

*    The Authors present this work as enabling collaboratively SSL in FL when clients are of different compute capabilities. However, only two types of networks ResNet-18/34 are considered and both train with batch size 200. That leaves the setting to just two types of clients. If the Authors considered a few more models to cover a wider spectrum of system capabilities (specially towards more lightweight models -- e.g mobilenet, efficientnet or similar?) and different batch sizes (which directly impacts on memory peak at training time), then I would be convinced that Hetero-SSFL is indeed addressing the client device heterogeneity problem.

*    The RAD is one of the key ingredients in Hetero-SSFL but the Authors do not provide details about it. Is it also a vision dataset? which one? How does the quality of the SSL change with different RADs?

*    The Authors state that the RAD "doesn't posses any special properties", could this be elaborated? I'm inclined to size and diversity in the RAD would be important, but to what extent? is MNIST a good RAD?

*    How the SSL training on the clients happens is clear. However, how is the linear evaluation protocol implemented in this work? Is there labelled "validation" set to train the classifier and then the evaluation is done on a, yet another, sub-set of data that the client has (i.e. a labelled test set)? Do these validation and test sets follow the same label distribution as the (unlabelled) training set each client has?

In Table 3, the Authors present some results on how Hetere-SSFL extrapolates from the cross-silo (5 total clients and 5 clients per round -- as considered for Table1 and, I believe, Table 2 also) to a cross-device setting. I understand the cross-device setting is more challenging, specially when it comes to ensuring the alignment of representations helps the training process.
*    Could the Authors comment upon why SSL suffers such a sharp drop in performance in cross-device settings?
*    It would be good to include in Table 3 the standard (non-SSL) FL baseline, which should be something in the high 50%s, given that FedAdam(https://arxiv.org/abs/2003.00295) achieves 52.5% in a setup with 500 clients (sampling 10 per round I believe). Adding that extra row is good for putting the results in context: SSL is really important and clearly the way to go for FL, but we aren't there yet.

**Summary Of The Paper:**

This paper presents Hetero-SSFL, a framework for SSL in FL that enables diverse clients (heterogeneous in terms of their compute capabilities and/or data distribution) to collaboratively learn a good SSL model. At the end of N rounds, the SSL model gets personalised (via linear evaluation protocol) to each client's data. Different from the majority of previous FL works, is the fact that clients in Hetero-SSFL do not communicate their locally-trained model with the server. Instead, what clients send are the embeddings outputted by the local model when using a common (i.e. same in all clients) dataset. These representations are then aggregated in the server and used in the following round to align (via a distance loss term) the representations each client learns during SSL. Hetero-SSFL is primarily evaluated on vision classification tasks but also text classification.

**Summary Of The Review:**

I am happy with this work but there are some open questions that need to be addressed ( see both the Strengths and Weaknesses lists) specially in regards to the RAD and results in Table1&2. I think the Authors can address these during the rebuttal period easily.

---

> ### Author Response · Authors · 2022-11-17
> **Response to Reviewer N6gv**
>
> We thank the reviewer for providing detailed feedback and helping us improve the paper. We are happy to see that the reviewer thinks that we address a known problem in an original way. We address reviewer's concerns below -
>
> 1. The linear evaluation protocol requires training of linear classifiers on top of the learnt encoders. For CIFAR-100, the evaluation in the IID case is done by learning a linear classifier for classification on a  dataset with 100 classes whereas in the non-IID setting the evaluation is over a 20-class dataset. Thus, even though the self-supervised models learnt in the IID case are much better, the difficulty of the end-task, used solely for the evaluation purpose, makes the performance metric slightly lower. This trend is also seen in the results reported in the prior works corresponding to FedU and FedEMA.
>
> 2. We experimented with more diverse types of networks like CNN and MLP to demonstrate the effectiveness of the approach and added the results in the Appendix subsection F.1. The section reads "To demonstrate the effectiveness of the framework with diverse types of heterogeneous clients, we experiment by adding clients with models of much smaller capacities like CNN and MLP along with Resnet18 and Resnet34 models. We use Hetero-SSFL in this setup on CIFAR-100 dataset in the non-IID setting to achieve federated training of self-supervised models on CIFAR-100 dataset and use linear evaluation protocol, as described in Section 5.1 of the paper to measure the test accuracy. We show the average performance of the small capacity clients, and average performance of the large capacity clients in settings with 20% and 40% small capacity models in Table below as compared to the standalone training"
> | Setting | Small Models | Resnet Models |
> | -------   | ----------------| -----------------|
> | 20% small | 39.4 $\pm$ 1.55 | 64.2 $\pm$ 2.9 |
> | 40% small | 37.5 $\pm$ 1.96 | 63.7 $\pm$ 1.89 |
> | Standalone Training | 28.8 $\pm$  0.9 | 55.3 $\pm$ 1.3 |
>
> 3. We have added a section, Section D Details About RAD, in the Appendix for providing details about the common public dataset aka RAD. A subset of the section reads "The Representation Alignment Dataset (RAD) is important for training in Hetero-SSFL. Functionally, the RAD is used to guide the simultaneous training on the peer clients by serving as a common dataset for representation alignment. The only assumption about this dataset is to be of the same domain as the training data, i.e., for image classification tasks it could be images sourced from the web and for the text-classification tasks it could be publicly available text like from Wikipedia, Reddit etc. In the experiments described in Table 1 and Table 2 of the paper, we subset a part of the dataset (5000 data points) and set it aside to be used as RAD before beginning the training process. This dataset is created before the clients are intialised and data is partitioned into clients. We further analysed the effect of using simple MNIST dataset as an RAD and saw similar results. The change in performance with respect to the size of RAD is given in the figure attached. When the RAD size is very low, the local models' performance is closer to the performance in standalone training. As the size of the RAD grows, the performance gets better but because the gain is small after 3000 points, we keep $L = 3000$ for all the experiment simulations." Link to the figure - https://imgur.com/Q5bNBhk
>
> 4. At each client, the dataset is partitioned into a train, validation and a test set. The train set is used for the training of the self-supervised model in federated learning using Hetero-SSFL while validation and test datasets are kept aside during this process. After the self-supervised encoder models are trained, we train a linear classifier to evaluate the performance of the trained self-supervised models. These secondary linear models are trained on the held out validation set and the performance metrics are finally reported over the test set.
>
> 5. Due to compute resource constraints for performing experiments, we sample a small subset ($0.1\%$) of all the local clients for local updates in each global iteration of the procedure. Thus, as the number of participating clients increase the number of updates on each client gets reduced. The performance gap becomes visible because the total number of model updates over the entire run become small as well as the main idea of Hetero-SSFL, learning from peer clients via representation alignment gets restricted due to partial participation. But even in this case, Hetero-SSFL is better than the baselines. And if we select more percentage of clients per global round, we are able to achieve even better performance.
>
> 6. We added a supervised upper bound in Table 3 of the paper corresponding to the FedAvg algorithm.

---

> > ### Author Response · Authors · 2022-11-21
> > **Regarding our response**
> >
> > Dear Reviewer N6gv,
> >
> > We have addressed your concerns in our responses. We would like to hear your feedback. Please feel free to raise questions if you have other concerns.
> >
> > Best regards,
> >
> > Authors

---

> > ### Comment · Reviewer_N6gv · 2022-11-24
> > **post-rebuttal feedback**
> >
> > I appreciate the time the Authors spent working on the rebuttal. I'll address is point in the same order:
> >
> > 1. Thanks for clarifying. I see these details were provided in Appendix C (but in Sec 5 this appendix was only highlighted as containing additional experiment). In my opinion these details regarding the experimental evaluation should be in the main paper. For example you can squeeze them in the `implementation details` or in `evaluation` paragraphs in `Sec 5.1`. FL is getting quite diverse these days, and because of this results should be provided in a way that are self contained (e.g. could captions in table 1&2 provide the most important details regarding the experiment?). I mention this because if someone is super familiar with SSL in FL and then checks the results of these tables, the reader will never suspect that evaluation protocol is done with just 2 classes for C10 and 20 for C100.
> >
> > 2. Thanks for running this experiment. Could the Authors provide further insights of what the small models exactly are? One way of doing so could be by expanding Table 8 and add info such as: number of parameters, FLOPs. That would be more informative regarding the memory/compute footprints of these models than "20% small"  or "40% small".
> >
> > 3. The new section about RAD makes the paper way more complete. Are the results shown in Figure3 done with MNIST as RAD? if not, could the Authors provide figures of what was the impact between going from CIFAR-100 RAD to MNIST (or any other non-CIFAR set of images)?
> >
> > 4. Thanks for clarifying.
> >
> > 5. Sampling 10% is very standard in cross-device evaluations for CIFAR-10/100 (although in real-world settings you'd normally sample even less). I find the cross-device results (right-most column in table 3) the weak point of Hetero-SSFL. Could the Authors speculate what would need to be done to prevent such a sharp drop in performance? (which makes it closely comparable to FedEMA). Maybe the aggregation methodology needs to be revised when having more clients?
> >
> > 6. Good!

---

> > > ### Author Response · Authors · 2022-11-29
> > > **Author response to pos-rebuttal comments**
> > >
> > > We thank the reviewer for continuously providing valuable feedback.
> > >
> > > We have addressed additional concerns below -
> > >
> > > 1. We thank the reviewer for highlighting this, we indeed want to re-structure and move some more details about the RAD and evaluation in the paper but have kept them in the Appendix for the rebuttal version due to space management during this limited amount of time.
> > >
> > > 2. In this experiment, in addition to the current models which include Resnet18 and Resnet34, two other models were added to the list of models each client can have, these two models were a 2 convolutional layered CNN and a MLP. The reported results are for CIFAR-100 dataset when 20% or 40% of the clients were restricted to have one of the smaller models. Below we provide details of the smaller models -
> > >
> > > | Model type | # params | # GFLOPs |
> > > | ------------| -------------| ---------- |
> > > | CNN | ~4M | ~10M  |
> > > | MLP | ~540k | ~7M |
> > > | Resnet18 | ~11.7M | ~17.8M |
> > > | Resnet34 | ~34.2M | ~37.7M |
> > >
> > > 3. The results shown in Figure 3 are not with MNIST dataset, they are with the held out dataset from CIFAR-100. If we use MNIST as RAD we obtain a performance of 63% $\pm$ 1.1 as opposed to 65% $\pm$ 0.9 for the experiment setting with local models of classes Resnet18 and Resnet34.
> > >
> > > 4. The aggregated representations, $\bar{\Phi}$, represent the current state of global knowledge and the effect of collaboration on local learning of a client is governed by the hyperparameter $\mu$, we think that in cross-device settings (with sampling), these two aspects could be looked into more deeply. Specifically, we could study (1) how much the current state of global knowledge $\bar{\Phi}$, should depend on the recently sampled clients vs clients which were inactive in last few iterations, this will be done by setting the weight vector **w** appropriately, and (2) if we need to vary $\mu$ with time independently for each client (and not dependent on the global iteration number) since each client's learning might be at different stages for each global round.

---

### Official Review · Reviewer_GXVZ · 2022-10-24

**Confidence:** 4
**Correctness:** 3
**Technical Novelty And Significance:** 3
**Empirical Novelty And Significance:** 3
**Recommendation:** 5

**Clarity, Quality, Novelty And Reproducibility:**

The construction of RAD is not clearly described. Otherwise, the paper is easy to read, and motivation is intuitive.

**Strength And Weaknesses:**

**strength**
- propose a new method to transfer the knowledge across clients without direct weight aggregation, so that allows participation in heterogeneous self-supervised models
- validate the model over text- and image-based classification problems
- provide the convergence analysis for the proposed model

**weakness**
- Missing communication cost analysis.
- Missing analyses of the RAD size and the construction of the RAD. To avoid the data privacy issue, the distribution of the RAD should be different from the distributions of local datasets (i.e., Non-IID). Therefore, the construction of the RAD can highly impact the model performance.
- Only two ResNet-variants are used for architecture heterogeneity. More diverse architectures are required for backbones, such as Transformers, simple CNNs, and MLPs.
- Performance gap with baselines decreases for larger participating clients at each round.

----
+ local clients from baselines do not train on the samples constructed to RAD?

**Summary Of The Paper:**

The paper proposes a new method for federated self-supervised learning with clients who may train on local tasks using heterogeneous model architectures. The authors adopt a Siamese structure for local self-supervised learning, implemented to BYOL or SimSiam, and validate it using both image and text classification tasks. To allow semantic aggregation of local clients' representation, they introduce a public dataset named Representation Alignment Dataset (RAD) that all local models utilize for implicitly transferring representations across clients by minimizing the CKA distance of RAD representation. Therefore, the proposed model doesn't need direct weight aggregation and allows communication with heterogeneous clients.


**Summary Of The Review:**

The paper proposes a practical framework allowing model heterogeneity during federated self-supervised learning. The method is reasonable and provides convergence analysis, but there is a lack of analyses to understand the behavior of the proposed methods and the issue of RAD construction.

---

> ### Author Response · Authors · 2022-11-17
> **Response to Reviewer GXVZ**
>
> We thank the reviewer for the valuable feedback and we are happy to see that the reviewer thinks that our solution is practical and useful. We address reviewer's concerns below -
>
> 1. We added a section in the Appendix, Section E Communication Cost Analysis, regarding the communication cost analysis. A subset of the section reads - "In Hetero-SSFL, each global round involves the server sending the common dataset, RAD, to each client and the clients sending back the representations obtained on the RAD. The size of the RAD is given by $L$, thus the communication cost in each round is $O(L)$. In all our experiments, we set the RAD size to be $3000$. Hence, the communication cost in each round of Hetero-SSFL corresponds to transferring $3000$ parameters twice. This is orders of magnitude less than the conventional FL algorithms that require sharing of the model parameters which run in millions. For federated training of the Resnet18 models, having 11M number of parameters, Hetero-SSFL with $L = 5000$ will require 2000 times less communication cost in each round making it much less communication intensive."
>
> 2. The similarity (or difference) in the RAD data distribution and local data distribution does not affect the training performance as RAD is a common dataset only used for aligning representations across clients. Moreover, since the local data distributions are not known outside the client, it is practically not possible to compare the client data distribution with any other data. We have added a section, Section D Details About RAD, in the Appendix for providing more details about the common public dataset aka RAD. A subset of the section reads "Functionally, the RAD is used to guide the simultaneous training on the peer clients by serving as a common dataset for representation alignment. Also, since the models are not transmitted and only the RAD representations are transmitted from clients to the server, there is no privacy loss due to involvement of RAD in the training process. The only assumption about this dataset is to be of the same domain as the training data, i.e., for image classification tasks it could be images sourced from the web and for the text-classification tasks it could be publicly available text like from Wikipedia, Reddit etc. Because the only requirement for RAD is to come from the same domain, it does not pose practical constraints. Several applications can make use of publicly available datasets to use for training in Hetero-SSFL, for example, learning self-supervised models for image datasets could use MNIST or CIFAR datasets for training, similarly acoustic applications can use audio datasets provided by Huggingface, and likewise."
>
> 3. We have experimented with more diverse types of networks like CNN and MLP to demonstrate the effectiveness of the approach and added the results in the Appendix. The section reads "To demonstrate the effectiveness of the framework with diverse types of heterogeneous clients, we experiment by adding clients with models of much smaller capacities like CNN and MLP along with Resnet18 and Resnet34 models. We use Hetero-SSFL in this setup on CIFAR-100 dataset in non-IID setting to achieve federated training of self-supervised models on CIFAR-100 dataset and use linear evaluation protocol, as described in Section 5.1 of the paper to measure the test accuracy. We show the average performance of the small capacity clients, and average performance of the large capacity clients in settings with 20% and 40% small capacity models in table below as compared to the standalone training."
>
> | Setting | Small Models | Resnet Models |
> | -------   | ----------------| -----------------|
> | 20% small | 39.4 $\pm$ 1.55 | 64.2 $\pm$ 2.9 |
> | 40% small | 37.5 $\pm$ 1.96 | 63.7 $\pm$ 1.89 |
> | Standalone Training | 28.8 $\pm$  0.9 | 55.3 $\pm$ 1.3 |
>
> 4. Due to compute resource constraints for performing experiments, we sample a small subset ($0.1\%$) of all the local clients for local updates in each global iteration of the procedure. Thus, as the number of participating clients increase the number of updates on each client gets reduced. The performance gap becomes visible because the total number of model updates over the entire run become small as well as the main idea of Hetero-SSFL, learning from peer clients via representation alignment gets restricted due to partial participation. But even in this case, Hetero-SSFL is better than the baselines. And if we select more percentage of clients per global round, we are able to achieve even better performance.
>
> 5. The local clients in the baseline do not use RAD for training. For Hetero-SSFL too, the RAD is used just used for aligning the representations across clients. In cases when $\mu=0$, the performance of the Hetero-SSFL is same as the performance of the clients with standalone training which ensures that the performance gain is not due to additional data (in the form of RAD).

---

> > ### Author Response · Authors · 2022-11-21
> > **Regarding our response**
> >
> > Dear Reviewer GXVZ,
> >
> > We have addressed your concerns in our responses. We would like to hear your feedback. Please feel free to raise questions if you have other concerns.
> >
> > Best regards,
> >
> > Authors

---

> > ### Author Response · Authors · 2022-11-29
> > **Look forward to your feedback.**
> >
> > Dear Reviewer GXVZ,
> >
> > We have addressed your concerns in our responses. We would like to hear your feedback. Please feel free to raise questions if you have other concerns.
> >
> > Thanks and regards,
> >
> > Authors

---

### Official Review · Reviewer_9iyo · 2022-10-26

**Confidence:** 4
**Correctness:** 3
**Technical Novelty And Significance:** 2
**Empirical Novelty And Significance:** 2
**Recommendation:** 3

**Clarity, Quality, Novelty And Reproducibility:**

Clarity: okay
Quality:  the paper written is not well-prepared and it requires proofread
Novelty: limited as core parts are similar to previous work
Reproducibility: okay

**Strength And Weaknesses:**

Pros: Overall, the idea of allowing different network architecture across clients in FL for SSL is very interesting. The selected technique foundation (Makhija et al., 2022) is reasonable.

Cons:
1.Limited methodology novelty. Despite of the different learning paradigm, Hetero-SSL is very similar to Makhija et al., 2022 (for supervised learning tasks). The core techniques discussed in Sec 3, including general formulation, RAD, CKA, etc., seems directly borrow from Makhija et al., 2022. The authors may want to highlight the differences if there is any or give sufficient credits to the previous work.

2.The assumption of requiring a common public dataset available in the server will raise many practical concerns. For example, what kind of data can be considered as the common dataset, how many data points need to be used, is it easy to find the privacy-free public data to serve this purpose, etc.

3.What do a and b mean in Assumption 4.1. I assume they are two iterations. Then don’t you want a,b > 0?

4.Lemma 4.5 is similar to Tan et al.  The authors should give proper reference here.

5.Are the results in Table 3 from IID or non-IID setting?

6.How to select the common (global) data is missing in Section 5.

7.Grammar and format issues should be corrected.


**Summary Of The Paper:**

Summary: This work proposes an FL paradigm for SSL under client heterogeneity. The proposed method, Hetero-SSFL allows each client to train personalized and different self-supervised models. It then enables joint learning across clients by aligning the lower dimensional representation with those of a common dataset assumed to be available on the server. The work is an extension of Makhija et al., 2022 by changing the supervised learning task to SSL. Also, Herero-SSFL has the impractical assumption of having a common dataset on the server, which may limit its application in some real settings. The author compared Hetero-SSFL with other FL+SSL methods on benchmark datasets.

**Summary Of The Review:**

The paper aims to address the interesting question, but the novelty and the written quality downgrade it from the other submissions.

---

> ### Author Response · Authors · 2022-11-17
> **Response to Reviewer 9iyo**
>
> We thank the reviewer for the helpful feedback, and it is encouraging for us to see that the reviewer finds the problem statement being tackled interesting. We address reviewer's comments below -
>
> 1. We agree that the framework for federation is similar to Makhija et al., 2022 and as suggested by the reviewer we modified the paper to give more credit to the previous work. However, we focus on the problem of self-supervised learning as opposed to supervised learning in this work which can ameliorate the problem of lack of labelled data on clients. The key takeaway and idea behind this work is showing that joint learning of the *task-independent representations* across clients is useful in federated learning, which is substantially different from the one in the previous work. We also prove convergence bound for general non-convex objectives for our method which is new as well. The design and training of efficient local models in self-supervised way is also a more difficult task than training supervised learning models, for example, for the NLP task, we learn transformer based models using MLM loss over the entire corpora. Furthermore, the evaluation of the trained self-supervised models require robust design as there are no predefined end-tasks. To this end, we provide extensive evaluation of the proposed method under multiple evaluation protocols like linear evaluation and semi-supervised learning protocol, under various federated learning settings in varied heterogeneous scenarios.
>
> 2. We have added a section, Section D Details About RAD, in the Appendix for providing details about the common alignment dataset aka RAD. A subset of the section reads "The Representation Alignment Dataset (RAD) is important for training in Hetero-SSFL. Functionally, the RAD is used to guide the simultaneous training on the peer clients by serving as a common dataset for representation alignment. The only assumption about this dataset is to be of the same domain as the training data, i.e., for image classification tasks it could be images sourced from the web and for the text-classification tasks it could be publicly available text like from Wikipedia, Reddit etc. Because the only requirement for RAD is to come from the same domain, it does not pose practical constraints. Several applications can make use of publicly available datasets to use for training in Hetero-SSFL, for example, learning self-supervised models for image datasets could use MNIST or CIFAR datasets for training, similarly acoustic applications can use audio datasets provided by Huggingface, for example."
>
> 3. In Assumption 4.1, a and b indeed represent iteration number and we have modified the paper to include a,b $\geq$ 0. We thank the reviewer for pointing it out.
>
> 4. We did modify the paper to more explicitly credit the prior work.
>
> 5. The results in the Table 3 are for non-IID setting and we have added that in the caption for the table.
>
> 6. We have added a section, Section D Details About RAD, in the Appendix for providing details about the common public dataset aka RAD. A subset of the section reads "In the experiments described in Table 1 and Table 2 of the paper, we subset a part of the dataset (5000 data points) and set it aside to be used as RAD before beginning the training process. Since the RAD is considered generic with no bearing on clients, it is created before the clients are intialised and data is partitioned into clients. We further analysed the effect of using simple MNIST dataset as an RAD and saw similar results."
>
> 7. We thoroughly proof-read the paper and corrected some minor issues. Since other reviewers did not raise issues with the writing, if the reviewer could point out specific issues we would be happy to address them.

---

> > ### Author Response · Authors · 2022-11-21
> > **Regarding Our Response**
> >
> > Dear Reviewer 9iyo,
> >
> > We have addressed your concerns in our responses. We would like to hear your feedback. Please feel free to raise questions if you have other concerns.
> >
> > Best regards,
> >
> > Authors

---

> > ### Comment · Reviewer_9iyo · 2022-11-24
> > **Thanks for clarifying the questions and adding the missing details.**
> >
> > Thanks for clarifying the questions and adding the missing details.
> >
> > ----
> > My major concern about not properly pointing out the similarity to the previous work still remains. Although this paper discusses SSL in FL, the contribution (in my view) is incremental to Makhija et al., 2022.
> >
> > First, the general formulation of this paper (Eq 1) is the same as Eq 3 of Makhija et al., 2022. This paper's Eq 2 is the same as that of Makhija et al., 2022. So for problem formulation, this paper seems to replace the supervised loss of Makhija et al., 2022 with a self-supervised loss. Even in the revision, such similarity was not addressed.
> >
> > Second, this work uses the same metric, Linear CKA, as Makhija et al., 2022 does for measuring the similarity of representation between clients (see Eq 4 of this paper and Eq 6 of Makhija et al., 2022).
> >
> > Third, the theoretical analysis does not show the SSL component. The assumptions are commonly used in the existing optimization literature and not specifically for SSL. The proof does not explicitly show how SSL played a role in the derivation. Please correct me if I am wrong.
> >
> > Overall, the method section gave me the impression of "copying" from Makhija et al., 2022. Regardless of the fact that the objective functions for supervised learning and self-supervised learning are different, I am afraid that it is not obvious to see the fundamental differences or challenges of adapting Makhija et al., 2022, to SSL settings. Besides, for ethical reasons, the authors might want to point out the similarity in the formulation and overall idea.
> >
> > -----
> > A common concern shared by all the reviewers is about RAD, which plays a very important role in the proposed method. The assumption that RAD is public and comes from the same domain is strong. A key application of FL is privacy-sensitive data analysis. Thus, a representative public dataset can be difficult to acquire. In addition, in heterogeneous settings, data from different clients can come from different domains. Such assumptions may limit the practical value of the proposed method. Also, this RAD needs to communicate with clients. Thus, communication costs need to be highlighted. It will be helpful to highlight the number of RAD data points required. As shown in the new results of Appendix Figure 3, if RAD data size =500, the performance of the proposed method is worse than the basline FedEMA.
> >
> > ----
> > Some minor comments: I also noticed that not all the edits after revision were highlighted (e.g., Assumption 4.1). Regarding grammar issues, this is really minor and won't affect my scoring. For example, it may be more proper to say "achieving a global objective" for the last sentence on page 3.
> >
> > ----
> > I agree with the contribution of exploring architecture-agnostic FL for SSL. However, the presentation of the paper should be improved by highlighting its novelty, providing sufficient references and details, and conducting further evaluation.

---

> > > ### Author Response · Authors · 2022-11-29
> > > **Author response to post-rebuttal comments**
> > >
> > > Dear Reviewer 9iyo,
> > >
> > > Thank you for the suggestions regarding the presentation. We would incorporate these changes in the next revision of the paper.
> > >
> > > Additionally, the details regarding the communication cost, selection and size of RAD, effect of the hyper-parameter $\mu$ have been added to rebuttal version of the paper.
> > >
> > > Best regards,
> > >
> > > Authors

---

### Decision · Program_Chairs · 2023-01-20

**Decision:**

Reject

**Justification For Why Not Higher Score:**

According to my expertise and reviewing process, this paper should belong to a Reject.

**Justification For Why Not Lower Score:**

According to my expertise and reviewing process, this paper should belong to a Reject.

**Metareview: Summary, Strengths And Weaknesses:**

This paper describes an FL paradigm for SSL under client heterogeneity. The proposed method, Hetero-SSFL allows each client to train personalized and different self-supervised models. It then enables joint learning across clients by aligning the lower dimensional representation with those of a common dataset assumed to be available on the server. The work is an extension of Makhija et al., 2022 by changing the supervised learning task to SSL. Also, Herero-SSFL has the impractical assumption of having a common dataset on the server, which may limit its application in some real settings. The author compared Hetero-SSFL with other FL+SSL methods on benchmark datasets. Overall, the idea of allowing different network architecture across clients in FL for SSL is very interesting. The selected technique foundation is reasonable.

However, there are several obvious weakness: 1) Limited methodology novelty. Despite of the different learning paradigm, Hetero-SSL is very similar to Makhija et al., 2022 (for supervised learning tasks). The core techniques discussed in Sec 3 seems directly borrow from Makhija et al., 2022. The authors may want to highlight the differences if there is any or give sufficient credits to the previous work. Although this paper discusses SSL in FL, the contribution (in my view) is incremental to Makhija et al., 2022. 2) The assumption of requiring a common public dataset available in the server will raise many practical concerns. For example, what kind of data can be considered as the common dataset, how many data points need to be used, is it easy to find the privacy-free public data to serve this purpose, etc. 3) A common concern shared by all the reviewers is about RAD, which plays a very important role in the proposed method. The assumption that RAD is public and comes from the same domain is strong. A key application of FL is privacy-sensitive data analysis. Thus, a representative public dataset can be difficult to acquire. Also, this RAD needs to communicate with clients. Thus, communication costs need to be highlighted. It will be helpful to highlight the number of RAD data points required. 4) The theoretical analysis does not show the SSL component. The assumptions are commonly used in the existing optimization literature and not specifically for SSL. The proof does not explicitly show how SSL played a role in the derivation. Overall, this paper may not be ready for publication at ICLR. The next version must be a strong paper if authors can take comments into consideration.